# Adapted Deep Embeddings: A Synthesis of Methods for *k*-Shot Inductive Transfer Learning

**Tyler R. Scott, Karl Ridgeway, Michael C. Mozer**
Department of Computer Science
University of Colorado, Boulder
{tysc7237,karl.ridgeway,mozer}@colorado.edu

## Abstract

The focus in machine learning has branched beyond training classifiers on a single task to investigating how previously acquired knowledge in a source domain can be leveraged to facilitate learning in a related target domain, known as *inductive transfer learning*. Three active lines of research have independently explored transfer learning using neural networks. In *weight transfer*, a model trained on the source domain is used as an initialization point for a network to be trained on the target domain. In *deep metric learning*, the source domain is used to construct an embedding that captures class structure in both the source and target domains. In *few-shot learning*, the focus is on generalizing well in the target domain based on a limited number of labeled examples. We compare state-of-the-art methods from these three paradigms and also explore hybrid *adapted-embedding* methods that use limited target-domain data to fine tune embeddings constructed from source-domain data. We conduct a systematic comparison of methods in a variety of domains, varying the number of labeled instances available in the target domain ($k$), as well as the number of target-domain classes. We reach three principal conclusions: (1) Deep embeddings are far superior, compared to weight transfer, as a starting point for inter-domain transfer or model re-use (2) Our hybrid methods robustly outperform every few-shot learning and every deep metric learning method previously proposed, with a mean error reduction of 34% over state-of-the-art. (3) Among loss functions for discovering embeddings, the histogram loss (Ustinova & Lempitsky, 2016) is most robust. We hope our results will motivate a unification of research in weight transfer, deep metric learning, and few-shot learning.

## 1 Introduction

Since the introduction of backpropagation, researchers in neural networks have investigated *inductive transfer learning* [3, 24]. Inductive transfer learning refers to the use of labeled data from a *source* domain to improve generalization accuracy on a related *target* domain with limited labeled data [23]. The notion of 'related' is not formally defined, though the existence of shared features across domains is presumed. With the deep learning movement, there has been a resurgence of interest in inductive transfer learning (*ITL*) for classification, which we will refer to as $k$-ITL, where $k$ denotes the number of labeled examples available for each class in the target domain. Of particular interest has been the case with small $k$, due to the fact that deep learning is typically data hungry, in contrast to human learners who often generalize well from a single example [16].

Three independent lines of research have tackled the $k$-ITL problem, either explicitly or implicitly. First, the *deep metric learning* literature [2, 4, 18, 20, 21, 26, 28, 32, 34, 35, 38] uses the source domain to construct a nonlinear embedding in which instances of the same class are clustered together and well separated from instances of different classes. The quality of an embedding is evaluated by

examining inter-class separation in the target domain. Because the target domain is just a means of evaluation, deep-metric learning is agnostic as to $k$. Second, the *few-shot learning* literature [8, 10, 12, 14, 25, 29, 31, 33] addresses the case when $k$ is small, typically $k \leq 20$. Many of these methods construct embeddings, just as in the metric-learning literature, though other methods have been explored, e.g., meta-learning. Third, there has long been an intuitive appeal to the *weight transfer* framework [1, 19, 24, 36, 37], which involves using the hidden representations obtained by training on the source domain as an initialization point for a second network to be trained on the target domain. In weight transfer experiments, large $k$ ($\geq 100$ or $\geq 1000$) are typically chosen.

Despite distinctive foci on $k$, all three lines of research utilize essentially the same architectures. They differ in two aspects of training: (1) the proposed loss function, and (2) whether weights are fine tuned on the target domain (which we refer to as *adaptation*). In this work, we compare state-of-the-art methods from each paradigm on a range of data sets, varying both the number of examples provided for each class in the target domain, $k$, and the number of classes in the target domain, $n$. We also formulate hybrid methods combining ideas across paradigms. We reach three strong conclusions:

- Weight transfer is the least effective method for $k$-ITL. For small $k$, the other methods yield vastly superior results; for large $k$, transferring weights from source to target domains yields little or no improvement over training from scratch on the target domain. This result has strong implications for the field: many researchers use weight transfer as a means of bootstrapping training in a novel domain, e.g., by starting with a state-of-the-art model such as VGG or AlexNet. Indeed, the TensorFlow development team has released a library of pretrained models, called TensorFlow Hub [30], specifically for this purpose. Our results indicate that this hub would better serve the community by providing pretrained embeddings.

- Across existing methods in few-shot learning and deep metric learning that discover embeddings, one specific loss function is most effective for small-$k$ ITL, the *histogram loss* [32]. This loss comes from the deep metric learning literature, and it has never previously been compared to losses from the few-shot learning literature.

- We propose a hybrid approach, *adapted embeddings*, that combines loss functions for deep embeddings with weight adaptation in the target domain. This hybrid approach robustly outperforms *every few-shot learning and every deep metric learning method previously proposed* on $k$-ITL. The performance differences are not in tiny percentage error reductions that distinguish contemporary methods, but are systematic and meaningful: a mean error reduction of 34% over state-of-the-art. To our knowledge, the only previous work to explore such a hybrid approach did so in a cursory manner and the results were ambiguous [33].

In the next section, we survey the three paradigms for $k$-ITL and identify a state-of-the-art method within each. Where multiple methods are roughly comparable in performance, we select based on simplicity of the method. We then describe an experimental methodology for systematically comparing methods, which includes the hybrid we propose, on a range of common data sets.

## 2 Paradigms for *k*-Shot Inductive Transfer Learning

### 2.1 Deep Metric Learning

An *embedding* is a distributed representation that captures class structure via metric properties of the embedding space. In *deep metric learning*, a neural network is trained to map from the input to the embedding space.[1] Various objective functions have been proposed for deep metric learning, all of which aim to ensure that instances of the same class are near one another in the embedding space and instances of different classes are far apart [2, 4, 18, 20, 21, 26, 28, 32, 34, 35, 38]. The objective functions differ in how they quantify 'near' and 'far'. Because classes are separated in the embedding, metric learning supports categorization of an unlabeled instance by projecting it to the embedding space and considering its proximity to labeled instances. Given a pretrained deep embedding, one can perform $k$-shot learning by embedding the $k$ instances of each novel class and then classifying unlabeled instances by their proximity to the labeled data.

Deep metric learning methods are evaluated using a variation of $k$-shot learning in which a *support set* of $k$ examples of $n$ classes is embedded, and a mean Recall@$r$ score is obtained for a *query set*, held-out examples of this domain. Recall@1 is simply nearest-neighbor classification and this single best guess is typically how $k$-ITL is scored. Although the entire range of $r$ is swept in evaluation, ranking of the methods is fairly consistent across $r$. Since there is not an emphasis on learning from few examples, $k$ typically varies in magnitude and is generally not directly specified.

The *histogram loss* [32], hereafter HISTLOSS, is a state-of-the-art method that we chose to represent the deep metric learning paradigm. Its Recall@1 performance is equivalent to or slightly better than contemporaneous methods [26, 34, 35], and HISTLOSS has only one hyperparameter, the number of histogram bins, and results are robust to the setting of the hyperparameter.[2] HISTLOSS constructs two sets of similarities, $S^+ = \{s(f_\phi(\boldsymbol{x}_i), f_\phi(\boldsymbol{x}_j))|y_i = y_j\}$ and $S^- = \{s(f_\phi(\boldsymbol{x}_i), f_\phi(\boldsymbol{x}_j))|y_i \neq y_j\}$, where $f_\phi(\boldsymbol{x}_i)$ is the neural network embedding of input $i$ with class label $y_i$ and $s(.,.)$ is a similarity metric. A loss, $\mathcal{L}_\phi = \mathbb{E}_{s\sim p^-}[\int_{-\infty}^s p^+(z)dz]$, is defined on the similarity distributions of positive pairs and negative pairs, $p^+(s)$ and $p^-(s)$ respectively. The distributions are each estimated as a histogram, and the empirical loss is efficiently computed using the histogram bins to identify all $(s^+ \in S^+, s^- \in S^-)$ similarity pairs for which $s^- \geq s^+$. The loss is minimized via stochastic gradient descent in weights $\phi$.

## 2.2 Few-Shot Learning

The few-shot learning literature is explicitly directed at the $k$-ITL problem with an emphasis on small $k$, typically $k \leq 20$. Embeddings form the basis of some methods [8, 12, 14, 29, 31, 33]. Meta-learning [10, 25] is another innovative approach involving training a recurrent network on a sequence of small classification tasks, so that it learns more efficiently on a subsequent task. We chose the *prototypical network* [29], hereafter PROTONET, as our representative of few-shot learning methods. It is simple and elegant, in addition to being state-of-the-art.[3]

PROTONET is a deep network that embeds input $\boldsymbol{x}_i$, and for each class $c$, a prototype $\boldsymbol{\mu}_c$ is constructed from the $k$ instances in the support set: $\boldsymbol{\mu}_c = \frac{1}{k} \sum_{\{i|y_i=c\}} f_\phi(\boldsymbol{x}_i)$. A query $q$ is classified according to its distance to the prototypes: $p(y_q = c|\boldsymbol{x}_q) \sim \exp(-d(f_\phi(\boldsymbol{x}_q), \boldsymbol{\mu}_c))$. The network parameters, $\phi$, are trained to maximize the conditional likelihood, i.e., $\mathcal{L}_\phi = \sum_i \ln p(y_i|\boldsymbol{x}_i)$.

## 2.3 Weight Transfer

Weight transfer in neural networks [1, 19, 24, 36, 37] is an instance of a more general framework in which parameters of a machine-learning model trained on a source domain are applied to a target domain. In some situations, the source and target are trained simultaneously [3, 27]. Some of the literature on weight transfer appears under the heading of *domain adaptation* [22, 27], which is often treated as a synonym for transfer learning, though formally domain adaptation involves changing input distributions instead of output labels [6].

The most systematic and thorough analysis of weight transfer is the work of Yosinski et al. [36]. In this work, the source and target domains share a common layered feedforward architecture, which maps input $\boldsymbol{x}_i$ to internal state $f_\phi(\boldsymbol{x}_i)$ which is then mapped to domain-specific class probabilities via a softmax, $p(y|\boldsymbol{x_i}) \sim \exp(\boldsymbol{\omega} f_\phi(\boldsymbol{x}_i))$, where $\boldsymbol{\omega}$ is a set of domain-specific weights. Yosinski et al. transferred various portions of $\phi$, from only the first layer of weights to all layers, up to and including the penultimate layer. In addition, the copied weights were either clamped after transfer or were further adapted on the target task. Training on the source task and adaptation on the target aimed to maximize the conditional likelihood, $\mathcal{L}_{\phi,\boldsymbol{\omega}} = \sum_i \ln p(y_i|\boldsymbol{x_i})$. Yosinki et al. found that the best classification accuracy on the target domain is obtained when all network weights up to the penultimate layer are transferred and then adapted. We will refer to this state-of-the-art scheme as *weight adaptation*, or WEIGHTADAPT for short.

Yosinski et al. mainly focused on large $k$, $k > 1000$, and observed only a modest improvement in accuracy over the baseline condition of ignoring the source-domain data and training only on the target-domain data. Nonetheless, the notion of weight adaptation is extremely popular in deep learning because it can provide a large time savings over training models from scratch, and it may prevent overfitting when the target domain is data constrained [24, 36].

### 2.4 Adapted Embeddings

We have summarized two representative, state-of-the-art embedding methods: HISTLOSS and PRO-TONET. For both methods, model parameters are determined solely based on the source-domain data. The target-domain support set—the $k$ instances of each of the $n$ classes in the target domain—are used merely for comparison to query (to-be-classified) instances. In contrast, weight adaptation determines model parameters using both source *and* target domain data. We explore a straightforward hybrid, *adapted embeddings*, which unifies embedding methods and weight adaptation by using the target-domain support set for model-parameter adaptation. To the best of our knowledge this seemingly obvious idea has been incorporated into only one few-shot learning paradigm, *matching nets* [33], referred to as *fine tuning*, and is beneficial in one domain, harmful in another.[4] Perhaps the assumption in the few-shot literature has been that little value will be obtained from adaptation with small $k$; indeed, for most algorithms, the data are insufficient to permit adaptation with $k = 1$. In the deep metric learning literature, the target domain is considered as a means of evaluating embeddings, and thus optimizing performance in the target domain is not a focus of interest.

## 3 Methodology

We tested six methods: WEIGHTADAPT, HISTLOSS, PROTONET, ADAPTHISTLOSS, ADAPTPROTONET, and a non-transfer BASELINE that ignores the source domain and trains a classifier solely on the limited labeled data in the target domain. We systematically explored how methods perform as a function of $k$ and $n$ on four popular data sets: MNIST [17], Isolet [5], tinyImageNet [9], and Omniglot [16].

Previous research on deep-metric and few-shot learning has addressed problems in which the number of available classes in the source domain, $N_{src}$, is much larger than the number of classes to be discriminated in the target domain, $n$. Consequently, training on the source is divided into a series of episodes where $n$ classes are sampled from the $N_{src}$.[5] In contrast, weight transfer has chosen problems in which $N_{src} = n$ and the same $n$ classes are used across training episodes, for a relatively large $n$. We had hoped to independently vary $N_{src}$ and $n$ in our exploration, but combined with search over $k$, the space becomes too large. We therefore assumed $N_{src} = n$. This constraint helps balance task difficulty across $n$: increasing $n$ makes the target task harder but also provides more data for training in the source domain. As a result, our simulations do not reach ceiling performance, which can be a concern in few-shot learning. Another rationale for this decision is that many real-world $k$-ITL tasks provide a limited supply of source data, as well as target data. For example, in medical radiology, one might hope to use labeled wrist x-rays to support the classification of ankle x-rays. To obtain robust and generalizable results, we evaluated models over $n$ ranging from 5 to 1000.

Also in the interest of robustness, we opted for another difference in methodology from most previous research on deep-metric and few-shot learning. Previous research has typically trained a single source model and evaluated over many episodes of the target domain. Statistical inference from these data allow one to predict the ranking of methods for new samples of the target domain, but *not* for new samples of the source domain. Consequently, we ran multiple replications of each method for a given $k$ and $n$, and on each replication we drew a single sample of $n$ classes from both the source and target domains.[6] This approach is computation intensive, but if method $X$ consistently outranks method $Y$, it should do so for a new (related) target domain, as well as a new (related) source domain. We

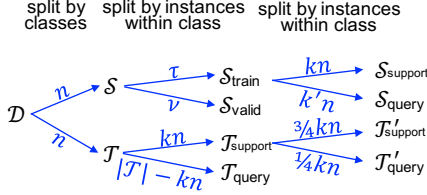

Figure 1: Data pipeline. Data set $\mathcal{D}$ is divided into source $\mathcal{S}$ and target $\mathcal{T}$ domains. $\mathcal{S}$ is further split into $\tau$ training and $\nu$ validation instances (see Table 1). From $\mathcal{T}$, $k$ support instances per class are selected and the rest become query instances. $\mathcal{T}_{\text{support}}$ is further split into support and query subsets for ADAPTPROTONET adaptation.

Table 1: Splits and sizes for each data set used in the $k$-ITL experiments. The source data set doesn't use a test split and the target data set doesn't use a validation split. The train size for the target data set, $\mathcal{T}_{\text{support}}$, is $k \times n$ for all data sets.

| | | Source Data Set | | Target Data Set | |
|---|---|---|---|---|---|
| | $n$ | Train Size ($\tau$) | Valid Size ($\nu$) | $k$ | Test Size |
| MNIST | 5 | $1600n$ | $600n$ | $\{1, 5, 10, 50, 100, 500, 1000\}$ | 10000 |
| Omniglot | $\{5, 10, 100, 1000\}$ | $15n$ | $5n$ | $\{1, 5, 10\}$ | $n(20 - k)$ |
| Isolet | $\{5, 10\}$ | $250n$ | $50n$ | $\{1, 10, 50, 100, 200\}$ | $n(297 - k)$ |
| tinyImageNet | $\{5, 10, 50\}$ | $350n$ | $200n$ | $\{1, 10, 50, 100, 300\}$ | $n(550 - k)$ |

expected to need many dozens of replications to obtain reliable estimates of mean performance, but to our surprise, we found that 10 replications was more than adequate to discern among methods.

All simulations were thus replicated 10 times. Each replication involved a random selection of classes and split of instances, as sketched in the data pipeline of Figure 1. To reduce variability, the same class and instance splits were used across methods, as were the contents of each minibatch of training data. Weights were initialized randomly for each replication.[7] For the source domain, a validation set was used to stop training. For target domain adaptation, training continued until performance reached asymptote. Given the small $k$ available for target domain adaptation, a validation set would have had high variance and the transfer of weights from the source should impose a strong inductive bias.

Table 1 contains details on the sizes and splits of each data set. The supplementary materials contain details on the network architectures used for each data set. For each data set, all six methods used the same underlying network architecture with two exceptions: (1) the BASELINE and WEIGHTADAPT architectures had an additional class-output layer which was not transferred from source to target; and (2) for training HISTLOSS and ADAPTHISTLOSS, the embeddings were L2 normalized, allowing for the use of the (bounded) cosine distance function with a 200-bin histogram. The embedding dimension was 128 for MNIST, Omniglot, and tinyImageNet, and 64 for Isolet. Because training parameters in PROTONET requires a data split between support and query sets, we chose to further divide $\mathcal{S}_{\text{train}}$ into $\mathcal{S}_{\text{support}}$ and $\mathcal{S}_{\text{query}}$ as noted in Figure 1. All models were trained with the Adam [13] optimizer.

## 4 Results

**MNIST**. This data set consists of $28 \times 28$ gray-scale images of handprinted digits [17]. MNIST was split into a source domain, with the digit classes 0–4, and a target domain, with 5–9. For this and following data sets, details of training parameters—learning rates and $k'$ (see Figure 1)—are included in the supplementary materials. Figure 2 plots accuracy on the test set, $\mathcal{T}_{\text{query}}$, for each of the six methods as a function of $k$, with $n = 5$ held constant. Each point is the average over ten replications. Error bands of $\pm 1$ standard error of the mean are shown, though they may be difficult to discern except when $k$ is small. The pattern of results here mirrors the results that we will present for the other data sets. Notably,

- WEIGHTADAPT shows modest improvements over BASELINE, but the benefit of the source domain diminishes as $k \rightarrow 1000$.
- For $k > 1$, ADAPTPROTONET improves on PROTONET, and ADAPTHISTLOSS improves on HISTLOSS. For $k = 1$, there are insufficient instances of each class to perform any adaptation, and thus the adapted algorithms are identical to their non-adapted counterparts.

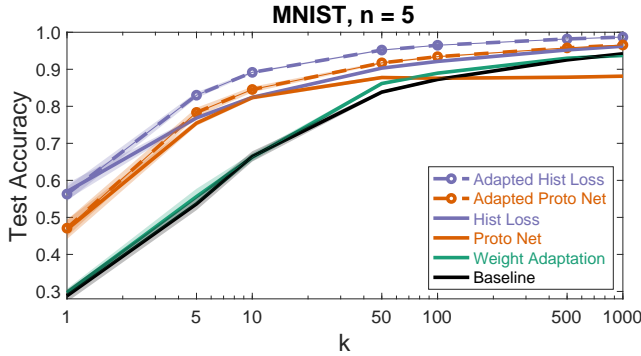

Figure 2: MNIST $k$-ITL results. Each point is the average test accuracy over 10 replications. Error bands indicate $\pm 1$ standard error of the mean.

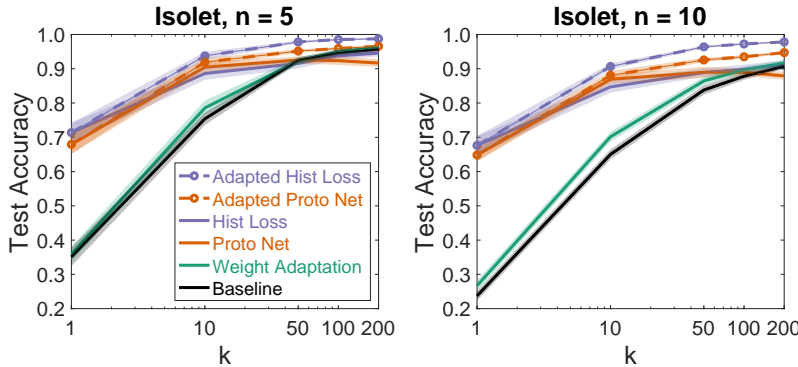

Figure 3: Isolet $k$-ITL results. Each point is the average test accuracy over 10 replications. Error bands indicate $\pm 1$ standard error of the mean.

- ADAPTHISTLOSS consistently outperforms ADAPTPROTONET.

- PROTONET appears not to benefit from $k > 50$, as one would expect for a method with high inductive bias which is designed for the small $k$ regime. However, ADAPTPROTONET continues to improve as more data are available because it can also use the data for adaptation.

- Across the range of $k$ tested, WEIGHTADAPT is inferior to the adapted embeddings, ADAPTHISTLOSS and ADAPTPROTONET.

**Isolet**. This data set, from the UCI repository, is a spoken letter (A-Z) data set with 26 classes and approximately 297 examples per class [5]. The input is coded as 617 attributes which specify spectral coefficients, contour features, sonorant features, pre-sonorant features, and post-sonorant features. The left and right panels of Figure 3 show test accuracy for $n = 5$ and $n = 10$, respectively. The results are qualitatively identical for the two values of $n$. The Isolet results eerily mirror those from MNIST (Figure 2), all the more surprising considering that the domains—vision and speech—and architectures—convolutional and fully-connected—are quite different.

**tinyImageNet**. This data set is a subset of ImageNet [7] containing 200 classes with 550 examples per class [9]. Each image is $64 \times 64$ with 3 channels for RGB. The few-shot literature typically uses miniImageNet for evaluation. We chose tinyImageNet because it has a greater diversity of classes (200 vs. 100). The three panels of Figure 4 show test accuracy for 5, 10, and 50-way classification problems. The take-away is similar to the previous two simulations, although WEIGHTADAPT does not seem to show as consistent a benefit over BASELINE as it did in the previous simulations. Once again, ADAPTHISTLOSS is consistently the best performer over all $(k, n)$ combinations.

**Omniglot**. This data set contains images of labeled, handwritten characters from diverse alphabets [16]. In the few-shot literature, Omniglot is the standard model-comparison data set. However, the literature relies on a specific split of the data on which state-of-the-art methods are now close to achieving ceiling performance. To avoid ceiling effects and obtain greater generality, we chose random splits. Omniglot has 1623 different characters, each with 20 instances; following previous research [29, 31, 33], we augment the data set with all $90°$ rotations, resulting in 6492 classes. Each grayscale image is resized to $28 \times 28$. The three panels in Figure 5 show test accuracy for 1-, 5-, and 10-shot learning. In each panel, $n$ is varied from 5 to 1000. Note that WEIGHTADAPT and BASELINE do not achieve performance much above chance for large $n$, and WEIGHTADAPT is reliably better than BASELINE for only $n = 5$. As in the previous simulations ADAPTHISTLOSS is robustly the

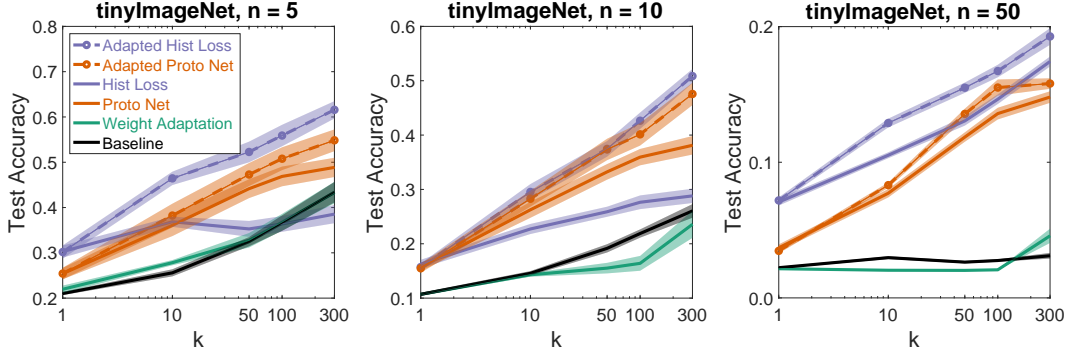

Figure 4: tinyImageNet $k$-ITL results. Each point is the average test accuracy over 10 replications. Error bands indicate $\pm 1$ standard error of the mean.

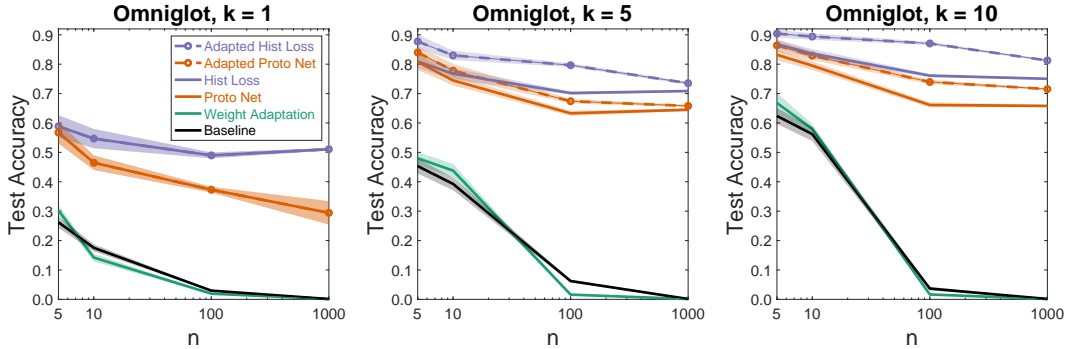

Figure 5: Omniglot $k$-ITL results. Each point is the average test accuracy over 10 replications. Error bands indicate $\pm 1$ standard error of the mean.

best performer, and the adapted embedding methods (ADAPTHISTLOSS, ADAPTPROTONET) reliably outperform the traditional embedding methods (HISTLOSS, PROTONET). (Remember that $k = 1$ does not provide sufficient data to permit adaptation.)

## 5   Discussion and Conclusions

The results from our $k$-ITL simulations are remarkably consistent across data sets and offer unambiguous prescriptions for significantly improving current practice in inductive transfer learning. The main messages are as follows.

**Adapted embeddings are the method of choice for $k$-ITL**. We proposed adapted-embedding methods, ADAPTHISTLOSS and ADAPTPROTONET, that combine deep embedding losses for training on the source domain with weight adaptation on the target domain. These methods are strictly superior to non-adapted (HISTLOSS, PROTONET) and non-embedding (WEIGHTADAPT, BASELINE) methods. Figure 6a summarizes 34 {data set, $k, n$} conditions by comparing the proportion reduction in classification error obtained by the best adapted embedding method (i.e., ADAPTPROTONET and ADAPTHISTLOSS) over the best of all alternative methods.[8] Figures 6b,c break the results down by comparing separately to non-adapted embeddings and adapted non-embedding methods, respectively. The adapted embeddings achieve an error reduction of 33.7% over the best of other methods, with a range from 2.2% to 73.9%. In every condition, adapted embeddings outperform non-adapted embeddings (mean 37.0%) and adapted non-embedding methods (mean 54.9%). Of the adapted embeddings, there is a clear ranking: ADAPTHISTLOSS is superior to ADAPTPROTONET.

To our knowledge, Vinyals et al. [33] is the only previous work to explore adapted embedding methods, in the context of matching networks. Few details were provided about the effort and the results were ambiguous. Several possibilities might explain why we see consistent and impressive

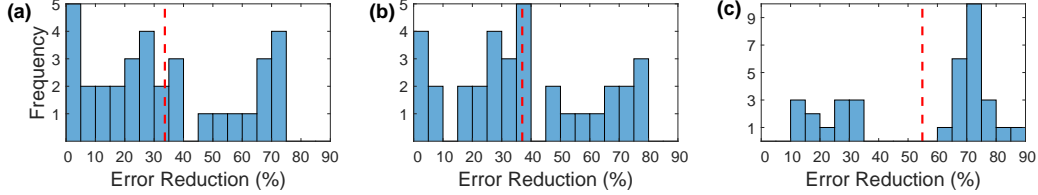

Figure 6: Histogram of percent reduction in classification error obtained by best adapted embedding method (ADAPTPROTONET, ADAPTHISTLOSS) versus the best of **(a)** all other methods, **(b)** non-adapted embeddings (PROTONET, HISTLOSS), and **(c)** adapted non-embedding methods (BASELINE, WEIGHTADAPT). Each histogram includes all of the 34 {data set, $k$, $n$} conditions tested with $k > 1$.

benefits of adaptation but Vinyals et al. did not. First, some algorithms appear to benefit more than others: for $k \in \{5, 10\}$, adapting HISTLOSS yields a greater benefit than adapting PROTONET. It's possible that matching nets overfit when adapting, whereas HISTLOSS, which has a natural stopping criterion, does not. Second, the evaluation of matching nets focused on $k = 1$ and $k = 5$. For $k = 1$, adaptation provides no information about intraclass structure; it can only separate classes. (And for the embedding losses we studied, we cannot do that with $k = 1$.)

**To construct models that can be repurposed, use deep embeddings.** WEIGHTADAPT is a common method of bootstrapping classifier training in a new domain. WEIGHTADAPT fails to match the adapted embeddings or even the non-adapted embeddings on $k$-ITL. WEIGHTADAPT does beat BASELINE for small $k$, but for our data sets, any advantage of WEIGHTADAPT seems to vanish for $k \geq 100$, in contrast to the adapted embedding methods that still benefit from increasing $k$. Our results are consistent with those of Yosinski et al. [36]. *TensorFlow Hub* and other libraries have been released to enable the reusability of large state-of-the-art models, in order to transfer and adapt their weights to novel target domains. Our results suggest that models trained on embedding losses would be far more accurate in transfer than models trained on an explicit classification loss, and should still achieve comparable training speed ups—one goal of model re-purposing.

WEIGHTADAPT decapitates a classification network and treats the penultimate layer as an embedding. So why does this embedding fail to be as useful for $k$-ITL as the embeddings discovered by PROTONET and HISTLOSS? The hidden layers of a classification network aim to discard information unrelated to class discrimination, and if successful, the penultimate layer will also orthogonalize the classes, i.e., discard most information about how one class relates to another. This inter-class structure is critical to projecting novel classes into an embedding space [26]. We thus argue that fundamentally, the objective—and the corresponding one-hot output representation of a classification network—is inferior for obtaining representations that will transfer to novel domains.

**Methods should not be segregated based on their focus on $k$.** Weight transfer, few-shot learning, and deep metric learning all perform a variant of $k$-ITL, yet these three lines of research have been mostly disconnected from one another. (For example, when submitting to NIPS, there are distinct subject areas for transfer learning, few-shot learning, and metric learning.) We suspect the lack of interaction is due to the fact that each paradigm has a distinctive focus on $k$. Although weight transfer may typically be used with larger $k$, our experiments show that it surprisingly beats BASELINE for small $k$. Few-shot learning is aimed at small $k$, but seems to work surprisingly well for large $k$. Metric learning is neutral as to $k$, but the representative method we chose, HISTLOSS, seems to work well for a range of $k$. If the preferred method depends on $k$, it might be sensible to treat these as independent topics, but one method—the hybrid ADAPTHISTLOSS—is superior for all $k$ and over a range of $n$.

The primary contribution of our work is the systematic comparison of methods across complementary lines of research. The novelty of ADAPTHISTLOSS—as a synthesis of HISTLOSS and WEIGHTADAPT—is admittedly minor: parameter fine tuning is a simple and obvious strategy in many areas of machine learning. What makes our work a valuable contribution is the non-obvious and impressive *magnitude* of improvements that are obtained by this obvious strategy. Many articles in metric learning and few-shot learning justify and differentiate methods based on tiny percentage error reductions, as contrasted with the comparatively impressive 34% error reduction we obtain over state of the art. By demonstrating gains of this magnitude, we hope to motivate a unification of research in weight transfer, few-shot learning, and deep metric learning.

## Acknowledgements

We would like to thank Chenhao Tan for helpful discussions. This research was supported by the National Science Foundation awards EHR-1631428 and SES-1461535.

The code is available at `https://github.com/tylersco/adapted_deep_embeddings`.

## Footnotes

[1]Deep metric learning methods are often initialized with a pretrained classification model such as AlexNet or VGG. One can decapitate its output layer and continue training with a metric-learning loss on the penultimate layer (e.g., [21, 32])

[2]To rank deep metric learning algorithms, we used comparisons directly reported in articles as well as performance on the same data sets and evaluation methodology. We obtain the partial ranking [32] $\geq$ [26, 34, 35] $>$ [4, 18, 21, 28, 38].

[3]Our partial ranking of few-shot learning methods based on target-domain accuracy is: [29] $>$ [10, 31] $>$ [25] $>$ [8] $>$ [12] $>$ [33] $>$ [14].

[4]The authors of [33] provide no details of how they fine tuned. They present results for fine tuning with $k = 1$, which cannot do much more than move all instances further apart.

[5]PROTONET [29] found advantages from sampling more than $n$ classes for source training episodes. 60-class episodes were constructed for source training on Omniglot. For miniImageNet source training, 30-class episodes were used when $k = 1$ and 20-class episodes were used when $k = 5$.

[6]In the supplementary materials, we show results from testing embeddings on the Omniglot data set using the methodology from previous few-shot learning studies.

[7]In [32], HISTLOSS was initialized with a pretrained classification model whose output layer had been decapitated, and training proceeded with the metric-learning loss. For the sake of comparison, we trained HISTLOSS from scratch.

[8]We exclude $k = 1$ conditions: one labeled example is insufficient to adapt either HISTLOSS or PROTONET.

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
