[Supplementary Material]

# Supplementary Materials for "Adapted Deep Embeddings: A Synthesis of Methods for $k$-Shot Inductive Transfer Learning"

## 1  Network Architectures in $k$-ITL Experiments

For each data set, all tested models used the same network architecture. Below are the details of these architectures:

**MNIST**  The MNIST architecture consisted of two convolutional layers, each with 32 filters, a $3 \times 3$ kernel, and a ReLU activation. The second convolutional layer was followed by a max-pooling layer with a $2 \times 2$ kernel and $2 \times 2$ stride, and finally, a fully-connected layer with 128 neurons.

BASELINE, WEIGHTADAPT, HISTLOSS, and ADAPTHISTLOSS used a learning rate of 0.005. PROTONET and ADAPTPROTONET used a learning rate of 0.001 and $k' = 100$.

**Isolet**  The Isolet architecture consisted of two fully-connected layers, the first with 128 neurons and a ReLU activation, and the second with 64 neurons.

BASELINE, WEIGHTADAPT, HISTLOSS, and ADAPTHISTLOSS are trained with a learning rate of 0.005. PROTONET and ADAPTPROTONET used a learning rate of 0.0001 and $k' = 50$.

**tinyImageNet**  The tinyImageNet architecture consisted of four convolutional layers, each with 32 filters and a $3 \times 3$ kernel, batch normalization, and a ReLU activation. The first three convolutional layers were followed by a max-pooling layer with a $2 \times 2$ kernel and stride. Following the four convolutional layers was a fully-connected layer with 128 neurons.

BASELINE, WEIGHTADAPT, HISTLOSS, and ADAPTHISTLOSS are trained with a learning rate of 0.005. PROTONET and ADAPTPROTONET used a learning rate of 0.0001 and $k' = 50$.

**Omniglot**  The Omniglot architecture consisted of three convolutional layers, all of which had 32 filters, a $3 \times 3$ kernel, a batch normalization layer, and finally a ReLU activation. The first two convolutional layers also had a max-pooling layer with a kernel and stride of $2 \times 2$ that followed the ReLU activation. The three convolutional layers were followed by a fully-connected layer with 128 neurons.

BASELINE, WEIGHTADAPT, HISTLOSS, and ADAPTHISTLOSS are trained with a learning rate of 0.005. PROTONET and ADAPTPROTONET used a learning rate of 0.0001 and $k' = 5$.

## 2 Embedding Results Using the Few-Shot Learning Methodology

The few-shot learning literature employs a training methodology in which the number of available classes in the source domain, $N_{src}$ is much larger than the number of classes to be discriminated in the target domain, $n$. Training on the source domain is divided into a series of episodes where $n$ classes are sampled from the $N_{src}$. We refer to this methodology as traditional or *episodic* training. Episodic training differs from the methodology in our $k$-ITL experiments in that we assume $N_{src} = n$, which we refer to as *restricted-source* training. (An additional difference exists in the testing procedure between our methodology and the traditional methodology, but this incidental difference should not affect results on expectation.)

Due to this difference in methodology, we evaluated HISTLOSS, ADAPTHISTLOSS, PROTONET, and ADAPTPROTONET on the Omniglot data set using episodic training, matching the traditional methodology. We omit BASELINE and WEIGHTADAPT since our experiments show they struggle to compete with the embedding approaches on the Omniglot data set and they do not easily lend themselves to an episodic training procedure.

Table 1: Embedding classification accuracies on the Omniglot data set using the few-shot learning training methodology. Standard error is reported over 1000 test episodes.

| | $(k, n)$ | | | |
|---|---|---|---|---|
| | $(1, 5)$ | $(5, 5)$ | $(1, 20)$ | $(5, 20)$ |
| HISTLOSS | $0.9864 \pm 0.0025$ | $0.9943 \pm 0.0013$ | $0.9461 \pm 0.0016$ | $0.9839 \pm 0.0011$ |
| ADAPTHISTLOSS | – | $0.9942 \pm 0.0013$ | – | $0.9797 \pm 0.0014$ |
| PROTONET | $0.9848 \pm 0.0034$ | $0.9960 \pm 0.0007$ | $0.9500 \pm 0.0027$ | $0.9864 \pm 0.0009$ |
| ADAPTPROTONET | – | $0.9950 \pm 0.0005$ | – | $0.9847 \pm 0.0011$ |

Table 1 presents transfer results on the Omniglot dataset using the traditional few-shot learning training methodology. Instead of sampling $n$ classes from the source domain for each episode, we sampled 60 classes, consistent with Snell et al. [1] and used the same data splits. The original data set with 90° rotations results in 6492 class, of which 4800 were selected for the source domain and the remaining 1692 for the target domain. Comparison of Table 1—episodic training—to the leftmost two panels of Figure 5 ($k = 1$ and $k = 5$)—restricted-source training—suggests that: (1) episodic training yields near-ceiling performance, which makes it impossible to meaningfully compare methods (which was our original motivation for restricted-source training); (2) with episodic training, the benefit of adaptation is unclear; (3) with episodic training, PROTONET may outperform HISTLOSS.

# References

[1] Snell, J., Swersky, K., and Zemel, R. (2017). Prototypical Networks for Few-shot Learning. In *Advances in Neural Information Processing Systems 31*, pages 4077–4087.