[Reviews · NeurIPS 2018]

Reviewer 1



I went back and read the main paper one more time. The following two points sums up the contribution of the paper. 1) L65-68: "We propose a hybrid approach, adapted embeddings, that combines loss functions for deep embeddings with weight adaptation in the target domain. This hybrid approach robustly outperforms every few-shot learning and every deep metric learning method previously proposed on k-ITL. " 2) L144-147: "In contrast, weight adaptation determines model parameters using both source and target domain data. We explore a straightforward hybrid, adapted embeddings, which unifies embedding methods and weight adaptation by using the target-domain support set for model-parameter adaptation" In plain English, this is just saying: "We use the test *and* train set to train embeddings in contrast to the standard practice of only using the train set" and it empirically worked slightly better. It's a no brainer that the performance increases as you also train on more (k) test data. Figure 2, 3 shows this (compare blue dotted line vs blue solid line). All in all, this is an experimental paper stating using additional examples from test domain increases the performance over not using them. Also, I think results on omniglot (smaller image; simple strokes on white background;) is very convincing in contrast to "online products" (larger images; real images on real background;) are anecdotal at best. For NIPS, I would not recommend the paper to be accepted, because the added value to the NIPS community is too limited. The described contribution is more technical rather than a big conceptual step. ------------------------------------------------------------------------------------------- This paper compares the lines of research on deep metric learning, few shot learning, and weight transfer and proposes hybrid approaches: weightadapt, adaptprotonet, adapthisloss. These hybrid approaches are obtained by deciding up to which layer you should initialize the network weights from other pretrained network and by choosing how to split the dataset (fig 1). The proposed hybrid approach is a mere hybrid and severely lack algorithmic novelty sufficient for a publication at NIPS. Furthermore, the experimental comparison is performed on simple datasets (MNIST, Isolet, tinyImagnet, Omniglot) which is not a fair comparison against Histogram loss [31]. In [31], the authors report results on CUB-200, Online Products which are significantly larger in size. Due to the lack of algorithmic novelty and inconclusive experiments, I vote for rejection.

Reviewer 2



The paper proposing to unite three distinct areas of transfer learning, namely, weight adaptation, few shot learning and metric learning. Weight adaptation is typically defined as fine-tuning a pre-trained network on a different but related problem. Few shot learning is defined as given a large labeled dataset, how can we further adapt when given a small set of examples for related but novel set of classes. Metric learning is defined as learning a metric space and typically tested in a few-shot retrieval setup in the literature. Hence, all these methods can be seen as handling different special case of same problem (called k-ITL in the paper). The paper experimentally shows that a trivial combination of two state-of-the-art method actually out perform all other methods for all three tasks. Hence, the main takeaway of the paper is necessity to consider these tasks in the same context and unite the literature. STRENGTHS The main claim of the paper is significant, clear, sensible and surprisingly was hidden in plain sight for so long. Clearly, metric learning and few-shot learning are closely related problems and they have been studied by different communities using similar techniques. Considering the way that metric learning is typically evaluated (Recall@R for novel classes), it is reasonable to apply metric learning to few shot learning problems. Somewhat surprisingly this combination results in state-of-the art method which outperforms existing few-shot learning algorithms. Although the algorithmic contribution is straightforward, the work will be significant to both metric learning and few-shot learning communities. The paper is written very clearly and it is easy to understand. Authors perform a thorough experimental study considering almost all factors. Although some factors like class imbalance in train and adaptation left unexplored, authors are upfront about them. WEAKNESSES I personally think the paper does not do justice to weight adaptation. The proposed setup is only valid for classification using same kind of data (modality, appearance etc.) between training and adaptation; however, weight adaptation is a simple method which be used for any problem regardless of change of appearance statistics or even changing the type of the modality. In practice, weight adaptation is proven to be useful in all these cases. For example, when transferred from ImageNet to audio spectrograms or transferred from classification to object detection, weight transfer stays useful. The paper studies a very specific case of transfer learning (k-ITL) and it shows that weight adaptation is not useful in this specific setting. Whether the weight adaptation useful in remaining settings or not still remains as an open question. The authors should clarify that there are more problems in transfer learning beyond k-ITL and they only consider this specific case. As a similar issue, the experimental setup only consider class distribution as a possible difference between training and adaptation. Clearly, low-level appearance statistics does not change in this case. It would be an interesting study to consider this since weight adaptation is shown to be handling it well. One simple experiment would be performing an experiment with training on tinyImageNet and adapting on CIFAR-10/100. This experiment would increase the robustness of the claims of the paper. Although it is little beyond of scope of k-ITL setup, I think it is necessary to be able to make the strong claim in L275-280. MINOR ISSUES: - Averaging the accuracies over different tasks in Figure 6 does not seem right to me since going from 90% to 95% accuracy and going from 10% to 15% should ideally be valued differently. Authors should try to give the same plot for each dataset separately in addition to combining datasets. - It is not direcly obvious that the penultimate layer will discard the inter-class relationship by orthogonalizing each class as stated in L281-L288. This statement is dependent on the type of the loss is used. For example, correctness of the statement is clear to me for L2 loss but not clear for cross-entropy. Either authors should provide some additional proof or justification that it is correct for all loss functions or should give more specifications. In summary, paper is stating an important but hidden in plain-sight fact that few-shot learning and metric learning is closely related and metric learning methods can very well be used for few-shot learning. This unified approach results in a straightforward algorithm which improves the state-of-the-art significantly. Although there are minor issues with the paper, it is an important contribution to both metric learning and few-shot learning communities. UPDATE: I have read the authors response and here are my comments on them: 1.a) Example citation for transfer from image to audio: Amiriparian et al., Snore Sound Classification using Image-Based Deep Spectrum Features. 1.b) Transfer beyond k-ITL: I also agree the proposed method should be able to handle this as well but unless we see an experiment on this, it is mostly a speculation. I think the other answers are satisfactory. I am keeping my score as 8.

Reviewer 3



The paper deals with the problem of transfer learning and tries to compare approaches under the seemingly different umbrellas of metric learning, transfer learning, and few-shot learning. It's a brilliant paper that makes the following empirically backed claims: (a) Weight adaptation is a surprisingly bad choice as it gets only marginal gains over training from scratch for a range of quantity of adaptation data available. (b) Methods trained on deep embedding losses in source domain do better at weight adaptation (AdaptHistLoss being proposed by authors) and outperform the previous SOTA by an average of 30% Some minor nits: (a) Line 104: "is is" -> "is" (b) Line 97: "has only one hyperparameter" -> state that the only hyperparameter is number of bins for completeness.